# Laboratory-Generated Autologous Skin Substitutes for Burn Treatment in Europe: Narrative Review, Experts’ Opinion, and Legal Considerations

**DOI:** 10.3390/ebj6020030

**Published:** 2025-06-03

**Authors:** Celine Auxenfans, Rocio G. Valencia, Philippe Abdel-Sayed, Miguel Alaminos, Jean-François Brunet, Fernando Campos, Jesus Chato-Astrain, Gloria Carmona, Anthony de Buys Roessingh, Stephanie Droz-Georget, Melinda Farkas, Ana Fernandez Gonzalez, Enikö Gönczi, Fredrik Huss, Bernd Hartmann, Barbara Heusi, Alexandra Karström, Naiem Moiemen, Giulia Sartoris, Antje Spranger, Marina Trouillas, Claudia Rosas, Jyrki Vuola, Vivienne Woodtli, Clemens Schiestl, Sophie Böttcher

**Affiliations:** 1Banque de Tissus et Cellules, Plateforme de Biotherapies, Hospices Civils de Lyon, 69002 Lyon, France; celine.auxenfans@chu-lyon.fr; 2Pediatric Burn Center, Children’s Skin Center, Department of Surgery, University Children’s Hospital Zurich, University of Zurich, Lenggstrassse 30, 8008 Zurich, Switzerland; rocio.valencia@kispi.uzh.ch (R.G.V.); melinda.farkas@kispi.uzh.ch (M.F.); enikoe.goenczi@kispi.uzh.ch (E.G.); barbara.heusi@kispi.uzh.ch (B.H.); giulia.sartoris@kispi.uzh.ch (G.S.); claudia.rosas@kispi.uzh.ch (C.R.); vivienne.woodtli@kispi.uzh.ch (V.W.);; 3Children’s Research Center (CRC), University Children’s Hospital Zurich, University of Zurich, Lenggstrasse 30, 8008 Zurich, Switzerland; 4Regenerative Therapy Unit, Service of Plastic, Reconstructive & Hand Surgery, Lausanne University Hospital, University of Lausanne, 1011 Lausanne, Switzerland; philippe.abdel-sayed@chuv.ch; 5Tissue Engineering Group, Department of Histology, Instituto de Investigación Biosanitaria ibs.GRANADA, University of Granada, 18012 Granada, Spain; malaminos@ugr.es.com (M.A.); f.campos@ugr.es (F.C.); jchato@ugr.es (J.C.-A.); 6Cell Production Center (CPC), Lausanne University Hospital (CHUV), 1005 Lausanne, Switzerland; jean-francois.brunet@chuv.ch; 7Andalusian Network for the Design and Translation of Advanced Therapies (AND&TAT)—Fundación Pública Andaluza Progreso y Salud, 41092 Sevilla, Spain; gloria.carmona@juntadeandalucia.es (G.C.); ana.fernandez.gonzalez@juantadeandalucia.es (A.F.G.); 8Children and Adolescent Surgery Service, Lausanne University Hospital, University of Lausanne, 1015 Lausanne, Switzerland; anthony.debuys-roessingh@chuv.ch; 9Laboratory of Stem Cell Dynamics, School of Life Sciences, École Polytechnique Fédérale de Lausanne, 1015 Lausanne, Switzerland; stephanie.droz-georget@chuv.ch; 10Department of Experimental Surgery, Lausanne University Hospital, 1005 Lausanne, Switzerland; 11Cell Production and Tissue Engineering Unit, Hospital Virgen de las Nieves, 18014 Granada, Spain; 12Department of Plastic and Maxillofacial Surgery, Department of Surgical Sciences, Uppsala University Hospital, 752 37 Uppsala, Sweden; fredrik.huss@akademiska.se (F.H.); alexandra.karstrom@akademiska.se (A.K.); 13Unfallkrankenhaus Berlin, 12683 Berlin, Germany; bernd.hartmann@ukb.de (B.H.); anja.spranger@ukb.de (A.S.); 14College of Medicine and Health, University of Birmingham, Edgbaston, Birmingham B15 2TT, UK; n.moiemen@bham.ac.uk; 15ATMP Unit, French Armed-Forces Blood Transfusion Center, 92140 Clamart, France; marina-laurie.trouillas@inserm.fr; 16Helsinki Burn Centre, Helsinki University Hospital, University of Helsinki, P.O. Box 263, 00029 Helsinki, Finland; jvuola@helsinki.fi

**Keywords:** autologous skin substitute, burn treatment, legal considerations

## Abstract

Autologous skin substitutes represent a promising advancement in the treatment of burn injuries, offering personalized solutions for patients with extensive skin loss. This white paper synthesizes the current knowledge on laboratory-generated autologous skin substitutes in Europe, incorporating expert opinions and legal considerations. The white paper examines the scientific principles underlying autologous skin substitute development, including cell sourcing, bioengineering techniques, and clinical applications. The regulatory framework governing the production and use of these advanced therapies in Europe is also examined, highlighting challenges in standardization, safety, and approval pathways. The text features expert insights that offer a real-world perspective on the clinical viability and translational hurdles of autologous skin substitutes. The findings highlight the potential of autologous skin substitutes to improve burn treatment outcomes while emphasizing the need for harmonized regulations to facilitate clinical implementation. Despite technological advancements, significant challenges persist, including production costs, scalability, and long-term efficacy. Another focus of this white paper are the legal changes, which have significantly impacted the production and availability of these technologies. The review concludes that while autologous skin substitutes hold great promise, further research, regulatory refinement, and interdisciplinary collaboration are essential to optimize their integration into clinical practice.

## 1. Introduction

One of the most important innovations in the treatment of severe burns in recent years have been ‘laboratory-generated autologous skin substitutes’. The possibility of producing sufficient autologous skin substitutes in the laboratory from a few centimeters of a patient’s skin biopsy within a few weeks is one of the most promising therapeutic approaches in the treatment of severe burns.

The European Skin Engineering Network (ESEN), initiated by the research committee as an official committee of the European Burns Association (EBA), has brought together experts from the various European burn centers and GMP (Good Manufacturing Practice) laboratories. Based on the high level of expertise of this group, this white paper attempts to provide an overview of the different autologous skin substitutes and their indications in the treatment of patients with severe burns in Europe.

Since 2009, autologous skin substitutes have been subjected to the regulatory requirements for Advanced Therapy Medical Products (ATMPs). We would like to outline for the reader how these regulatory changes have impacted the field—specifically, how the stricter requirements have negatively affected the availability and accessibility of these innovative cell therapies in European burn centers.

Finally, we will discuss possible approaches that we believe could help this promising therapy become more accessible to all burn patients in Europe. This includes a constructive dialogue with the legal authorities and a joint initiative of European burn centers to establish guidelines, outcome databases, and a structured training program for the use of autologous skin substitutes in all European burn centers. In order to achieve the above objectives, this initiative is fortunately supported by the European Union’s ‘COST Action’ program.

## 2. Different Types of Laboratory-Generated Autologous Skin Substitutes

Rupert E. Billingham, a biologist from the UK, performed a lot of research on the potential of transplanting sheets of pure epithelium and non-cultured epidermal cell suspensions for wound healing in guinea pigs in 1952 in Birmingham and London [1]. His actual research interest was in the basic research of organ transplantation. During the late 1970s and early 1980s, his method became the subject of research in the treatment of severe burns conducted by Howard Green and James G. Rheinwald in Boston [2,3]. Two decades later, Steven Boyce from Cincinnati was the first to develop a bi-layer autologous skin substitute composed of a matrix containing autologous fibroblasts and autologous keratinocytes placed on top of the matrix [4]. Since the initial application of laboratory-produced autologous skin substitutes, numerous subsequent variations and further developments have been employed globally and, in particular, across Europe.

### 2.1. Cultured Epithelial Autografts (CEAs)

Cultured epithelial autografts (CEAs) were the first product produced in the laboratory from a patient’s biopsy to treat massive burn injuries. The protocol for producing CEAs was developed in Boston at Brigham and Women’s Hospital by Rheinwald and Green [2,3,5,6,7], who managed to produce stratified autologous keratinocyte sheets. To do so, a small biopsy of the patient’s skin was taken. The epidermis was enzymatically processed to obtain keratinocytes, which were subsequently expanded in culture. After approximately 18–21 days, a stratified cell sheet developed. This thin layer of keratinocytes was transferred onto a Vaseline gauze and then applied topically on the patient’s debrided burn wounds. In the initial period following the successful clinical application of the CEAs, numerous centers in Europe (e.g., those in Lausanne, Lyon, and Turin) [8,9] sent their personnel to Boston to engage in collaborative learning with Howard Green and his team, seeking to gain insights into the production of CEAs.

While the fundamental principles of CEA production have been preserved, each center subsequently introduced specific modifications to the protocol, reflecting institutional expertise, local resources, and evolving regulatory frameworks.

Common Core Methodology

Across the centers in Lausanne, Lyon, and Paris, the core workflow for CEA production remains consistent and comprises the following steps:**Skin biopsy collection** from the patient, typically a split-thickness specimen;**Enzymatic digestion** to isolate keratinocytes from epidermal tissue;**Cell expansion** in culture, commonly using a feeder layer of fibroblasts;**Stratified sheet formation** over a culture period of approximately 10 to 21 days;**Harvesting and transfer** of the epithelial sheet onto a Vaseline gauze for clinical application or storage.


**Localized Variations in Standard Operating Procedures**



**Lausanne (Centre Hospitalier Universitaire Vaudois—CHUV) Figure 1**


Lausanne hosts the longest-running CEA program in Europe, with over 40 years of continuous operation. The CHUV adheres closely to the original Rheinwald and Green protocol. A 10 cm^2^ split-thickness biopsy is enzymatically digested using trypsin to extract keratinocytes, which are then seeded onto a feeder layer of Mitomycin C-inactivated 3T3 mouse fibroblasts. Following 18–21 days of culture, the resulting stratified keratinocyte sheets are transferred onto Vaseline gauze and packaged for clinical use [10]. Notably, the CHUV is currently undergoing the marketing authorization procedure (Authorisation de Mise sur le Marché, AMM) with Swissmedic, marking a significant step toward formal regulatory approval in Switzerland.


**Lyon**


The CEA protocol at the Lyon center, which also boasts over 35 years of experience [11], incorporates a two-step enzymatic digestion using dispase and trypsin for keratinocyte isolation. Expansion is carried out on irradiated human fibroblasts in a culture medium composed of Dulbecco’s Modified Eagle’s Medium (DMEM) and Ham’s F12, supplemented with fetal calf serum, hydrocortisone, insulin, isoprenaline hydrochloride, tri-iodothyronine, adenine, epidermal growth factor, and antibiotics. Keratinocyte sheets are generated within 10–15 days by seeding 8000–10,000 cells/cm^2^ in specialized culture flasks with peelable lids. The sheets can be used fresh or cryopreserved at −80 °C in a cryoprotective medium containing 10% dimethyl sulfoxide (DMSO) and 20% calf serum. In addition, Lyon’s facility is equipped to produce allogenic sheets derived from surgical discarded skin obtained from consenting donors undergoing procedures such as breast reduction or abdominoplasty.


**Paris**


In Paris, the production protocol remains closely aligned with the original Rheinwald and Green method. Keratinocytes are enzymatically isolated and expanded using a culture medium enriched with irradiated fetal bovine serum and irradiated human allogeneic dermal fibroblasts, which support the maintenance of clonogenic potential and stratification capacity. Paris has also pioneered the development of a human plasma-based epidermal substitute (hPBES). In this approach, an amotosalen-inactivated pool of fresh frozen donor plasma is used to generate a biological matrix. After keratinocyte passage, both irradiated fibroblasts and keratinocytes are seeded onto the matrix and cultured under immersion for 14 days [12,13]. The final product is overlaid with non-adherent gauze, packaged in transport containers, and delivered to the clinic for surgical application.

### 2.2. Co-Cultures of Autologous Keratinocytes and Fibroblasts (CDEAs)

Cultured dermal–epidermal autografts (CDEAs) represent an advancement over CEAs by incorporating both autologous cultured keratinocytes and fibroblasts, with relevant interactions between both cell types [14,15]. CDEAs were developed in the 1990s and have been used clinically, particularly for deep partial- and full-thickness burns. To create these constructs, a full-thickness skin biopsy is taken from the uninjured skin of the patient and subjected to enzymatic and mechanical treatments to isolate the necessary cell types. Co-culturing keratinocytes and fibroblasts is challenging due to their differing cell proliferation kinetics; fibroblasts multiply much more quickly than keratinocytes, inhibiting the linear growth of the latter. Consequently, for CDEAs, in vitro cultures of keratinocytes and fibroblasts are prepared separately, and then superimposed before being transplanted. After in vitro expansion and specific cell stimulation, the resulting bicomponent cellular sheets are applied to patient wounds similarly to CEAs. Thus, CDEA preparation is complex and requires long manufacturing times of 6–8 weeks compared to 3–4 weeks for CEAs. Therefore, the clinical use of CDEAs is limited to patients hospitalized for long time periods, such as patients with severe burns [8]. Nevertheless, once the initial culture period is completed, continuous production and the cryopreservation of patient-specific cells allow for a more flexible and on-demand availability of new cultured autografts. Despite the difficulty of preparation, the biphasic structure of CDEAs provides a more robust graft than CEAs for skin reconstruction, allowing for a lower contraction rate and making them suitable for areas where skin contraction must be minimized, such as joints and the neck [16].

### 2.3. Cell Suspension/Spray

In the mid–late-1990s, various research groups worked on techniques to transplant not cultured stratified epithelial sheets, but keratinocytes in suspension. The fundamental publications on this method were published by Björn G. Stark et al. [17] at the time. The rationale was that culturing stratified sheets was found to be complicated, labor-intensive, and time-consuming. A more straightforward way would be to just culture a large number of keratinocytes to be transplanted onto the wound surface, thus avoiding several steps in the process. Theoretically, by avoiding letting the keratinocytes confluence and stratify, in the culture they are kept in a proliferative state which could be positive when they are transplanted onto wounds. In the middle of the 2000s, in Perth, Australia, Fiona Wood and her team developed a method for spraying autologous keratinocytes onto wounds. This method was first used extensively in 2002 to treat the many severely burned victims of the Bali bombing [18].

There are two ways of using autologous cell sprays:A spray containing only cultured keratinocytes from the patient’s skin biopsy. Expansion of the cells is achieved basically according to the abovementioned techniques. However, the cells are passed before they reach confluence. Once enough cell number is reached, the keratinocytes are detached as above, mixed and washed, and centrifuged to a cell pellet. In the operating theater, the cell pellet is re-suspended in the thrombin part of a commercially available fibrinogen–thrombin tissue glue and sprayed onto the wound surface.A spray containing non-cultured cells, mainly keratinocytes, but also other cells such as melanocytes, which are isolated from a small biopsy and immediately processed into a spray directly in the operating theater and then returned to the patient. In this case, the spray functions as a tool to evenly distribute the cells on the wound surface, where they then start multiplying.

In Europe, the Berlin-Marzahn Hospital Burn Center (Unfallkrankenhaus Berlin), Germany, which is linked to the German Institute for Cells and Tissue (DIZG), has used autologous cultured keratinocyte sprays for many years. In fact, the DIZG produces autologous cultured keratinocyte sprays for all the German burn centers [19,20,21].

Another cell spray is produced in the laboratory of the Uppsala University Hospital in Sweden. They have transitioned to a xenofree approach, eliminating the need for a feeder-cell layer in their cell culture. The expanded keratinocytes are enzymatically detached and harvested into a cell suspension that is mixed with tissue glue and sprayed on the wound [22]. If compared to the production of CEAs, the spraying strategy demands less manual labor and thus, reduces the costs of production; however, lower take rates are often observed.

### 2.4. Autologous Two-Layer Skin Substitutes

There are two research groups in Europe that have followed the same path in the last 20 years and have been successful with their autologous bi-layer skin substitutes in the context of translational research.

Since 2012, the University of Granada, Spain, developed the University of Granada model (UGRSKIN) of bioengineered skin using nanostructured fibrin–agarose biomaterials containing keratinocytes and fibroblasts [23,24,25,26,27]. After preclinical studies, the ATMP product was translated into GMP regulation at the GMP facility of Hospital Virgen de las Nieves, Granada, coordinated by the Andalusian Network for the Design and Translation of Advanced Therapies (And&Tat-RAdytTA). This ATMP has obtained the hospital exemption authorization at the Hospital Virgen del Rocío from the Spanish Medicine Agency called ‘Piel humana obtenida por ingeniería de tejidos (PHIT)’ which is currently being used to treat severely burnt patients in the reference burn unit in Hospital Virgen del Rocío, Seville, with good clinical results.

To produce PHIT/UGRSKIN, a 9 cm^2^ skin biopsy is taken from the patient and processed to isolate fibroblasts and keratinocytes, which are cultured in their specific media. The cultured fibroblasts are incorporated into a fibrin–agarose matrix made of human plasma from blood donors (fibrin source) and commercial type VII agarose. The cultured keratinocytes are then seeded upon this matrix forming a dermo-epidermal construct. The process from biopsy to construction takes approximately 4–5 weeks. The dermo-epidermal construct is thereafter subjected to nanostructuring methods based on plastic compression and dehydration. Since the product was authorized as a GMP product, the bi-layer autologous skin substitute (PHIT/UGRSKIN) has already been used to cover extensive burn wounds in 19 patients at the burns center in Hospital Virgen del Rocío, Seville (18 after compassionate use authorization and one under hospital exemption authorization) [28,29].

In the mid-2000s, the research team at the Tissue Biology Research Unit began the preclinical testing of a bi-layer skin substitute in close cooperation with the Pediatric Burn Centre at the University Children’s Hospital in Zurich, Switzerland. This autologous bi-layer skin substitute consists of a mechanically compressed collagen hydrogel matrix containing the autologous fibroblasts and autologous keratinocytes on top of the matrix [30,31] (Figure 2). In the meantime, this bi-layer autologous skin substitute called denovoSkin^TM^ has been successfully applied in both a Phase 1 and a Phase 2b clinical trial and will be the subject of a Phase 3 clinical trial with several European burn centers as participating sites shortly [32,33].

It is interesting to note, and important for the discussion, that one part of the initial research team has since then continuously been working on obtaining the regulatory approval for denovoSkin^TM^. To do so, a spin-off company of the university was founded. To our knowledge, this is the only product (apart from the aforementioned CEAs from the CHUV team in Lausanne for Switzerland) for which marketing authorization could be granted in Europe in the foreseeable future.

In the meantime, the other part of the team, which remained at the University Children’s Hospital team (Skin and Soft Tissue Research Center, University Children’s Hospital Zurich—SSTaRC) is working on the integration of other cell types for an even more complex two-layer autologous skin substitute, including autologous melanocytes and endothelial cells.

## 3. Indications

Even if it must be assumed that laboratory-generated autologous skin substitutes can be helpful for any extent of burns, they are particularly important in cases where there is not enough donor area to harvest the needed amount of autologous split-thickness skin grafts to cover the wounds in time and to save the life of the patient. This lack of donor sites can exist from the time of admission (extensive burn injuries) or arise during the treatment of a severe burn injury, e.g., due to severe wound infections.

Close cooperation between plastic surgery and intensive care is essential in the day-to-day management of severely burned patients. Intensive care treatment for patients with severe burns has significantly improved in the last 50 years. Patients with a severe burn injury can now survive the first few days after the trauma [33,34]. However, to secure the lives of these patients, it is crucial to remove the deeply burned areas and cover them with autologous skin as soon as possible.

It is important to note that all of the methods described below require a bridging strategy. Ideally, allografts or dermal templates are used to provide competent wound coverage after very early excision of the necrotic tissue until the autologous skin substitute is available.

It is also important to note that only a very limited number of European burn centers are currently able to use these innovative autologous skin substitutes.

### 3.1. Indications for CEAs and CDEAs

In the first 10 years after the introduction of CEAs, the indication was generally for burns over 50% of the total body surface area (TBSA) [35,36,37,38,39]. In Europe, the indication of the affected body surface has often been determined by the accessibility and availability of CEAs. In France, even today, CEAs are only indicated for burns larger than 70% TBSA [40]. Subsequently, as it was not possible to achieve stable take rates, particularly in full-thickness burns, it was increasingly used in combination with wide-meshed autologous STSG. Another reason not to use CEAs alone to cover deep burns is that the resulting skin is often unstable and thus can significantly reduce the patient’s quality of life. In general, the use of CEAs is still indicated in patients with burns greater than 50% TBSA. However, for full-thickness burns, they should be used in combination with meshed split thickness skin grafts (STSGs) whenever possible, whereas deep dermal areas and donor sites can be covered with CEAs alone. The great advantage of CEAs is their relatively short production time of approximately 14–21 days in combination with their high expansion rate (1:1000 up to 1:3000) For the CDEAs, almost the same applies as for the CEAs, although the take rate can be marginally better. However, because of the addition of autologous fibroblasts to the product, production takes 28–42 days.

### 3.2. Indications for Cell Spray

The indication for a spray of cultured autologous keratinocytes is usually in combination with wide-meshed autologous STSGs. However, in contrast to sheet grafts, the major advantage of the spraying technique is that it can also be applied in combination with the MEEK micrografting technique [19,41,42]. In addition to this, like CEAs, the spray can be used to speed up the healing of donor sites after harvesting STSGs for extensive burns.

### 3.3. Indications for Autologous Two-Layer Skin Substitutes

Since the first application of bi-layer autologous skin substitutes from Steven Boyce’s laboratory towards the end of the 1990s, their use has not yet been established in burn treatment. Mainly due to the complex manufacturing process and the new legal regulations that have come into force in Europe in the meantime. At this point, it is worth emphasizing that the high regulatory requirements have driven up the cost of these innovative autologous products in particular. We will, of course, come back to this later. Currently, the indication for applications as experimental therapy is massive full-thickness burns of at least 80% TBSA. PHIT/UGRSKIN, recently authorized under Hospital Exemption, is indicated for both adults and children from one year of age at Hospital Virgen del Rocío, Seville. It is used for treating burns of various etiologies when no alternative treatment is available. However, the results achieved are so encouraging that there is great potential for their use in the future. As mentioned earlier, other products such as denovoSkin^TM^ are currently starting Phase III before regular application can be applied.

Notably, due to the long production time of 4–5 weeks and the simultaneous need for the early excision of the deep and extended burn wound, a so-called bridging strategy is unavoidable [43].

### 3.4. General Remarks on the Indication of All Types of Laboratory-Generated Autologous Skin Substitutes

Care should be taken to obtain the biopsy as early as possible to manufacture laboratory-generated autologous skin substitutes. The sooner the biopsy is taken after the admission of a patient with a severe burn injury, the better the quality of the biopsy will be. If different laboratory-produced autologous skin substitutes are used, a precise schedule is required. It is important to know what substitutes are available for the patient at what time. If a combination with autologous STSG is planned, sufficient donor sites must be available at that exact time.

The right ‘bridging strategy’ is important, especially for the grafting of bi-layer autologous skin substitutes, but also for CEAs and CDEAs. Modern burn surgery involves removing deep and partial deep burn areas as soon as possible to minimize the impact of necrosis on the whole organism. This is a race against time. Until the laboratory-generated skin substitute is ready for use, the wound must be covered to stabilize the patient, protected from infection complications, and, ideally, pre-treated (already forming a neodermis) before coverage with a laboratory-generated autologous skin substitute. For this reason, different types of allografts (glycerol preserved or fresh frozen), xenografts, and finally dermal templates (biological or synthetic) are of crucial importance. To ensure optimal timing for the patient between autograft, allograft, CEA, CDEA, and two-layered skin substitutes, collaboration between the production laboratory and the burn center is essential. This collaboration is also crucial to ensure proper follow-up and care for the grafted areas.

## 4. Regulatory Issues

ATMPs are innovative therapies that have created great hope for patients suffering from previously untreatable diseases and injuries [44]. In 2009, the European regulatory framework (Regulation (EC) No 1394/2007) took charge of the implementation of the new regulations for advanced therapies in Europe. The aim was to define standards for ATMPs and to make sure that all the patients in Europe have access to these new therapeutic options.

Since then, it has become clear that the production and use of CEAs or any other kinds of autologous skin substitutes for the treatment of severe burns all fall under the ATMP regulation and therefore, have to meet the standards defined by the new regulations of the European Union (Part IV of EudraLex—Volume 4) [45].

In addition, this means that the production of CEAs or any other laboratory-generated autologous skin substitute has to be carried out according to GMP in specialized facilities complying with the same requirements as any pharmaceutical product.

### 4.1. European Regulatory Issues Until 2009

About 20 years before the European regulatory framework was installed, from 1990 to 2009, the use of all kinds of laboratory-generated autologous skin substitutes was going to become a standard therapy in several highly equipped burn centers in Europe, supported by numerous publications. More and more laboratories capable of producing different types of laboratory-generated autologous skin substitutes were set up in Europe. Usually, these laboratories were attached to European burn centers—e.g., Vienna, Birmingham, Lyon, Berlin, Uppsala, etc. [46,47,48]. But there were also other laboratories like the one in Lausanne/Switzerland that served on a national basis for all three Swiss burn centers for the production and delivery of CEAs and CDEAs, as well as the laboratory in Asturias/Spain that provided skin products for most burn units in Spain [39,49].

The European Medicine Agency’s (EMA’s) guidelines, set forth under EMA/CAT/852602/2018, detail the necessity for convincing proof of concept in relevant animal models, safety evaluations, and appropriate administration protocols. Notably, skin products were already in clinical use long before these enhancements in regulatory expectations were implemented. Therefore, the requirements for non-clinical data became diverse with different EMA guidelines providing recommendations in order to characterize the type of product. However, the revolution of the variety of advanced therapy products is a challenge not only for scientists but also for regulators, so the assessment of the associated risks needs to be evaluated on a case-by-case basis according to the risks and the intended uses.

### 4.2. European Regulatory Issues After 2009

The regulatory landscape for ATMPs in Europe underwent significant changes following the implementation of Regulation (EC) No 1394/2007 in 2009. This regulation emphasized the necessity of thorough preclinical characterization and stringent quality control measures prior to the clinical application of ATMPs, mandating approval from National Medicine Agencies (NMAs) for clinical use. EMA provided the guidelines that most NMAs follow; however, the complexity of these regulations has posed considerable challenges, particularly for hospital-based laboratories producing autologous skin substitutes. One of the primary obstacles faced by laboratories is the requirement to comply with GMP standards, originally designed for large pharmaceutical manufacturers. Many hospital-based facilities lack the resources to meet these standards, which include the need for adequately trained personnel, suitable laboratory environments, a rigorous quality system, and the financial capacity to conduct all the necessary quality controls and validations. The high costs associated with GMP compliance further include raw material expenses and their control and continuous training obligations, hindering the pragmatic delivery of autologous skin substitutes to patients. The EMA’s guidelines, set forth under EMA/CAT/852602/2018, detail the necessity for convincing proof of concept in relevant animal models, safety evaluations, and appropriate administration protocols. Notably, skin products have already been in clinical use long before these enhancements in regulatory expectations. Therefore, the requirements for non-clinical data have become diverse and are typically evaluated on a case-by-case basis according to the associated risks and intended uses. Additionally, the ambiguity surrounding product requirement guidelines has left developers vulnerable to differing interpretations by NMAs, further complicating the regulatory process. There is a pressing need for enhanced communication among NMAs, researchers, and clinicians, and the creation of consensus documents could help standardize and optimize the production processes at the preclinical stage. Despite the intention behind the 2009 regulations, there has been no tangible progress in the use of autologous skin substitutes for the treatment of severe burns. As a result, many laboratories have stopped providing manufacturing services under the hospital exemption, as manufacturing under the new regulatory requirements has driven up costs enormously. The case of Cutiss Ltd. (Schlieren, Switzerland), which emerged as a successful spin-off company, showcases how regulatory hurdles can be navigated, yet the overall outlook for other, especially for academic, facilities is bleak. The high costs associated with compliance with the ATMP regulations have driven up the price of skin substitutes dramatically (approximately 10-fold). For instance, while the current costs for cultivating and delivering CEAs to severe burn patients are approximately EUR 7 per cm^2^, full compliance could inflate costs to around EUR 1 million for patients with large TBSA burns—a prohibitive expense [44] (Table 1).

## 5. Research and Future Perspectives

The future-oriented therapeutic option of a lifesaving wound coverage for a patient with a severe burn injury, which is independent of the available donor sites and at the same time superior to the current standard STSG method in terms of late results, requires a structured scientific approach.

### 5.1. Basic Research

In order to further develop the laboratory-produced autologous skin substitutes already in use, important efforts are also needed at the basic research level. In principle, there are two essential improvements:

First, adding to the complexity of the currently established products to get as close as possible to normal skin. The next steps are expected to be the integration of autologous melanocytes and endothelial cells in the near future. However, sensitive nerves and skin appendages such as sweat glands and hairs are also needed. It has been demonstrated that alternative non-skin cells, especially mesenchymal stem cells [50], or their acellular products can contribute to skin regeneration.

Second, reducing production time—again, basic research is crucial, especially as future autologous skin substitutes become more complex, containing multiple cell types.

### 5.2. Translational Research

The importance of translational research has increased enormously in the context of tissue engineering. A GMP laboratory is indispensable for all autologous products that are produced in the laboratory and need to be brought to the bedside. More and more burn units are seeking regular access to such a GMP laboratory. Hospitals must be prepared to fund these GMP laboratories since the costs of running such a laboratory are enormous. Synergies with other disciplines must be sought. The initiative of the Andalusian authorities to fund projects like the one in Granada is exemplary. After immense effort, a Hospital Exemption authorization was obtained, allowing production within the Andalusian Public Healthcare System at Hospital Virgen de las Nieves. The treatment is now available for use in Andalusia’s reference center for severe burns at Hospital Virgen del Rocío, Seville, all under the coordination of the Andalusian Network for the Design and Translation of Advanced Therapies [28].

One example of successful translational research is a variant of a two-layer autologous skin substitute that is pre-vascularised and contains melanocytes. The Pigmented–Vascularized Skin (PV-Skin), developed at the University Children’s Hospital/University of Zurich, Switzerland, is an autologous skin substitute produced from keratinocytes, fibroblasts, endothelial cells, and melanocytes [51,52,53,54].

### 5.3. Clinical Research

In the field of clinical research, outcome measurements are crucial for the future. Uniform outcome research in these patients has been hampered by the intermittent availability of autologous skin substitutes in Europe. Over the next few years, it is essential that prospective clinical studies are conducted to investigate how these products affect late outcomes as the various products move toward regulatory approval. It is important to objectively analyze and document the results after the application of these innovative methods, and therefore, more objective assessment tools need to be promoted.

## 6. Opportunities for Further Development in Europe

There is no doubt that the use of laboratory-generated autologous skin substitutes significantly improves the survival and quality of life of patients with severe burns. Unfortunately, the new regulations for ATMPs introduced in 2009 led to the shrinking of the availability of sufficient laboratories to meet the needs of European burn centers. We are still far away from a situation where every severely burned patient admitted to a European burn center has access to this innovative therapy. For this reason, the multi-professional ESEN team has developed three proposals, which we would like to share with you in this white paper:A constructive dialogue is needed with those officially responsible for the regulatory framework in Europe (EMA) and all the stakeholders including the users; producers, in particular the European burn centers; and most importantly, burn survivors. The aim of this constructive dialogue should be to define with users the minimum necessary regulatory requirements for laboratory-produced autologous skin substitutes as AMTPs, on the premise that these products have already been in use for over 40 years.Despite the regulatory requirements for patient safety, it would be important to allow European burn centers to set up a network of laboratories with affordable financial resources to ensure the provision of care to severely burned patients for all the European centers.For their part, European centers should work collaboratively towards establishing general guidelines and uniform outcome research for this innovative therapy.

Since its foundation, ESEN has set itself the goal of achieving these points.

A major success in the course of ESEN’s active collaboration was that the European Union approved an application for support and ESEN became a COST Action project in 2024.

The European Union-funded COST Action project called ESENBURN is made up of the following sub-projects (work packages):Regulatory requirements;Production conditions;Clinical application;Database.

‘COST Action’ is a funding organization for research and innovation networks across Europe and beyond and enables researchers and innovation to grow their ideas by sharing them with their peers, so there is the possibility that other interested parties can participate in such networks. This means that in addition to the ESEN team, more than 60 experts from all over Europe are participating in this network in order to boost research and innovation in the field.

## Figures and Tables

**Figure 1 ebj-06-00030-f001:**
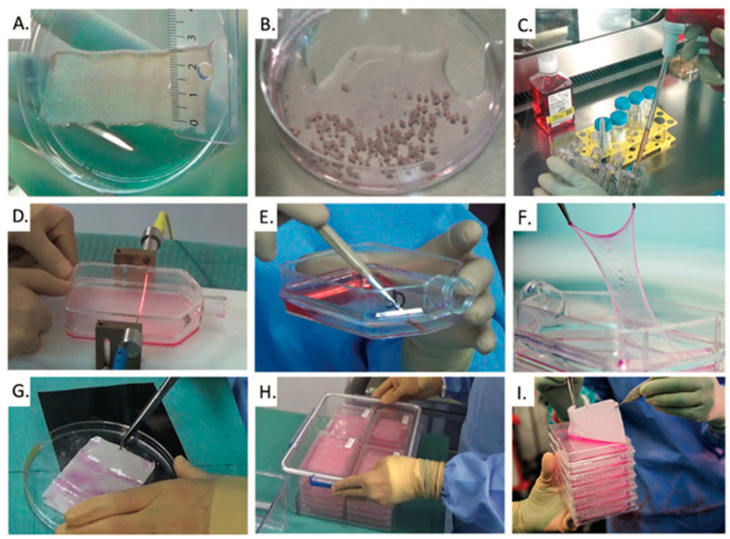
Various steps in the manufacturing of cultured epithelial autografts (CEAs): (**A**) A split-thickness skin biopsy of approx. 10 cm^2^ from a patient needing a skin substitute. (**B**) Keratinocytes are isolated by the enzymatic digestion of minced skin fragments. (**C**) The cells are expanded in culture, enzymatically harvested, and re-seeded in new culture flasks to expand cell number further. (**D**–**F**) After 18–21 days, the keratinocyte cultures form stratified sheets, which are enzymatically released and harvested from the culture flasks. In order to remove the keratinocyte sheets from the flask without disrupting them, one side of the flask has to be cut open. (**G**) The cell sheets are transferred onto a Vaseline gauze for increased stability and subsequently transferred to transport boxes containing a small quantity of cell culture medium. (**H**,**I**) Transport boxes are placed in a hermetically sealed secondary container for shipment to the clinic where the CEAs are applied to wounds in the operating theater [8].

**Figure 2 ebj-06-00030-f002:**
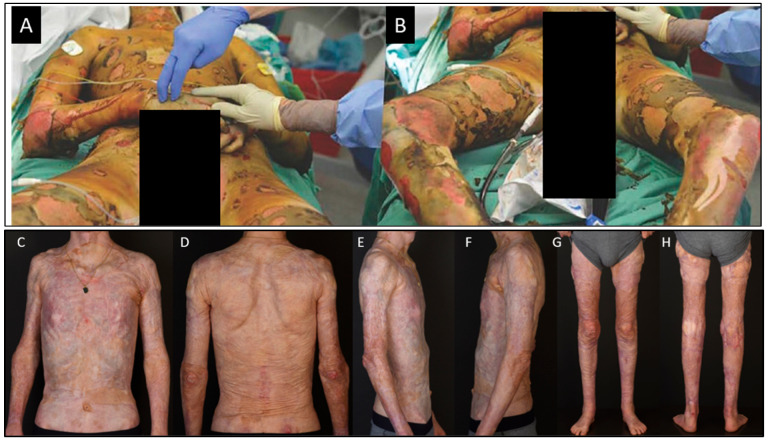
Massive burn injury with 95% TBSA deep burn; only a total of 600 cm^2^ of donor sites (parts of the scalp and both backs of the feet) was available for skin grafting 15,000 cm^2^ of burnt body surface. (**A**,**B**). The same patient 2 years after the trauma. Three different autologous skin substitutes were used: CEAs, CDEAs, and two-layer skin substitutes (**C**–**H**).

**Table 1 ebj-06-00030-t001:** Comparative overview of laboratory-generated autologous skin substitutes in Europe: clinical use, indications, production time and regulatory requirements. * Please note that these costs may vary from one European Country to the other.

Type of Skin Substitute	Clinical Use Since	Indication	Production Time	Estimated Cost	Regulatory Requirements
**Cultured Epithelial Autografts (CEAs)**	Early 1980s → since 2009 with national exemption authorisation	≥50% TBSA Deep dermal burns (2° deep) or 3° if needed; often combined with widely meshed STSG	10–21 days	7 EUR per cm^2^	- GMP production; - Preclinical data required; - Application more difficult and costly since 2009
**Cultured Dermal–Epidermal Autografts (CDEAs)**	2000s → since 2009 with national exemption authorisation	≥50% TBSA	28–42 days	10–12 EUR per cm^2^	- GMP standards, classified as ATMP; - Complex co-culture production requirements
**Cell Suspension/Spray (cultured)**	2000s → notably after Bali bombings 2002 → since 2009 with national exemption authorisation	50% TBSA, combined with: widely meshed STSG or MEEK	28–35 days	5–7 EUR per cm^2^	- GMP standards, classified as ATMP; - Complex co-culture production requirements
**Autologous two-layered Skin Substitutes**	2000 → only compassionate use	Experimental therapy for burns ≥ 80% TBSA; currently under compassionate use or hospital exemption	28–42 days	Not yet available *	- GMP production, extensive preclinical data; - Legal barriers prevent broad use and increase costs

## Data Availability

The original contributions presented in the study are included in the article.

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
