# Peer review of "Laboratory-Generated Autologous Skin Substitutes for Burn Treatment in Europe: Narrative Review, Experts’ Opinion, and Legal Considerations"

_2673-1991, 2025, doi:10.3390/ebj6020030_

Round 1
Reviewer 1 Report
Comments and Suggestions for Authors
This is a review paper on lab generated autologous skin substitutes with a focus on Europe. This paper could be published in its current format but it could be improved. The topics covered by the paper are too broad. For example it talks about ReCell autologous skin spray which is not a lab generated skin substitute. The treatment of ReCell is not thorough enough, however. There are almost no clinical data on efficacy or description of clinical trials. There is a good deal of information about the regulatory environment in Europe which may or may not be relevant to a clinical readership. A mention of PV Skin from Zurich towards the end of the paper seems out of place and seems too brief. Overall, I would focus the paper and add clinical trial data.
Author Response
Comment 1: This is a review paper on lab generated autologous skin substitutes with a focus on Europe. This paper could be published in its current format but it could be improved. The topics covered by the paper are too broad. For example it talks about ReCell autologous skin spray which is not a lab generated skin substitute. The treatment of ReCell is not thorough enough, however.
Author's Answer: If so, we will skip it
Comment 2: There are almost no clinical data on efficacy or description of clinical trials. There is a good deal of information about the regulatory environment in Europe which may or may not be relevant to a clinical readership. A mention of PV Skin from Zurich towards the end of the paper seems out of place and seems too brief. Overall, I would focus the paper and add clinical trial data.
Author's Answer: Thank you very much we tried our best
Reviewer 2 Report
Comments and Suggestions for Authors
This manuscript describes a review of the different approaches in Europe to treat burns with autologous skin substitutes.
Strength:
This is an interesting description of the advances made in patient treatment, given that Europe has been responsible for several advances in this field. The clarifications provided regarding the legal regulations relating to the transfer of these therapies to patients are relevant.
Weaknesses:
- There is heterogeneity in the descriptions of the different protocols, particularly in the level of details provided.For example, it would be clearer to describe the general protocol for CEA production, then briefly specify the specifics for each center (Lyon, Paris);The description of PHIT/UGRSKIN is very detailed, whereas denovoSkin is very brief.
- For each protocol, essential technical details such as culture time should be specified.It would be interesting to add a table summarizing the different therapies, total culture times, methods of use and indications (combined with mesh autograft, on donor site, on second-degree burns, third-degree burns, etc.), the outcome and possibly the cost of each therapy (per cm2).
- A clear description of the advantages and disadvantages of each therapy is missing; these could be added to the table, in addition to being detailed in the text.It would be very informative to specify the long-term healing after each treatment, and its impact on contracture, hypertrophic scar formation, etc.
- Page 9, line 346, we would like to know more about these so encouraging results.
- The regulatory issues section is relevant, but far too long and repetitive. It would have more impact if it were summarized in its main points (same for translational research). Please specify whether the therapies are and will be reimbursed by the local health care system.
- The future perspectives section could be further developed to better justify the most essential developments for future skin substitutes.
Author Response
Comment 1:
Strength: This is an interesting description of the advances made in patient treatment, given that Europe has been responsible for several advances in this field. The clarifications provided regarding the legal regulations relating to the transfer of these therapies to patients are relevant.
Weaknesses:
There is heterogeneity in the descriptions of the different protocols, particularly in the level of details provided.For example, it would be clearer to describe the general protocol for CEA production, then briefly specify the specifics for each center (Lyon, Paris).
Author's answer:
Thank you for the valuable advice, we have reorganised and reformulated the entire chapter 2.1.
Comment 2:
The description of PHIT/UGRSKIN is very detailed, whereas denovoSkin is very brief.
Author's answer:
We have summarize the PHIT/UGRSKIN according to your recommendations
Comment 3:
For each protocol, essential technical details such as culture time should be specified.It would be interesting to add a table summarizing the different therapies, total culture times, methods of use and indications (combined with mesh autograft, on donor site, on second-degree burns, third-degree burns, etc.), the outcome and possibly the cost of each therapy (per cm2).
Author's answer:
Thank you very much, we included a new table on page 10.
Comment 4:
A clear description of the advantages and disadvantages of each therapy is missing; these could be added to the table, in addition to being detailed in the text.It would be very informative to specify the long-term healing after each treatment, and its impact on contracture, hypertrophic scar formation, etc.
Author's answer:
Thank you very much. We changed it accordingly to your suggestions.
Comment 5:
Page 9, line 346, we would like to know more about these so encouraging results.
Author's answer:
Some of the promising results in treated patients have been published in the article 29. Martin-Piedra, M.A.; Carmona, G.; Campos, F.; Carriel, V.; Fernandez-Gonzalez, A.; Campos, A.; Cuende, N.; Garzon, I.; Gacto, P.; Alaminos, M. Histological assessment of nanostructured fibrin-agarose skin substitutes grafted in burnt patients. A time-course study. Bioeng Transl Med 2023, 8, e10572, doi:10.1002/btm2.10572. Additional clinical data are currently being collected in accordance with regulatory recommendations and are expected to be published in 2025.
Comment 6:
The regulatory issues section is relevant, but far too long and repetitive. It would have more impact if it were summarized in its main points (same for translational research). Please specify whether the therapies are and will be reimbursed by the local health care system.
Author's answer:
Thank you very much for this suggestion, we tired to shorten both.
Comment 7:
The future perspectives section could be further developed to better justify the most essential developments for future skin substitutes.
Author's answer:
Thank you very much, we rewrite this part.
Reviewer 3 Report
Comments and Suggestions for Authors
This paper aims to review multiple facets/aspects of the manufacture and use bioengineered skin substitutes in Europe. It would be improved by significant editing to focus on the problem at hand, which is the introduction of legislation that takes no account of products in use for many decades in several centres, and is severely restrictive to manufacturing and translational research in this area. Following this, there needs to be a succinct and comprehensive set of structured proposals to address these problems.
Line 51 What is a ‘white paper’ in this context?
Line 63 – what does ‘long term efficacy’ refer to? It is well established that autologous bioengineered skin products persist and mature after engraftment. Perhaps consider expanding this statement for clarity
Section 2.1 the authors could consider whether the outlines of manufacturing methods for CEAs usefully add anything to the paper. Same should be considered for the autologous 2 layer substitutes, unless the methods details are not available elsewhere.
Possibly a synthesis summarising the different products, and highlighting their differences and what the clinical and regulatory significance of these might be, would be more informative.
Line 211 what does ‘cells are passed’ mean?
Line 231: ‘might be at the cost…’ does not seem a very scientific way to describe anything
Line 270: elsewhere the locations of various institutes have been stated: where is the Tissue Biology Research Unit?
Line 297; while co-operation between ICU and plastic surgery (assuming plastic surgeons are the burns surgeons) is relevant, it might be better to put the consideration of ‘bridging strategies’’ here with a more detailed consideration of the key role wound preparation plays in the success of bioengineered skin substitutes.
(Section 3.4 – use of ‘you’ is not scientific writing)
Section 3.1 ‘Indication’ tends to be a clinical term, although local protocols for using these products might be determined by availability.
Line 319 – not clear what size of burn has to do with a change to use in conjunction with ssg
Line 323 - this statement should have a reference
Line 334 – reference for use and effectiveness with Meek and increased speed of healing
Line 339 – 2- layer skin substitutes – what is meant by ‘established’? CEA and cell sprays are by no means standard of care in many/most burns services
Line 374: this is the first time ATMP is spelt out but it occurs before this in the text
Section 4 Regulatory issues: this section (especially 4.1) should be in the background/introduction, as it really contains the justification of the need for this paper
Line 431: what does ‘diverse’ mean here and what is the significance of being evaluated on a case-by-case basis?
Line 440: why have hospitals stopped using the hospital exemption framework – does this exempt them from the GMP requirements?
Section 5.2 Perhaps consider the availability of prefabricated GMP clean rooms and their potential to decrease costs?
Circling back to talk about another product does not seem to add much to what is already a very broadly focussed paper..
Line 525 ESEN and other abbreviations should probably be spelt out the first time they occur, even though there is an abbreviation section at the end
Line 545: ‘Cost action’ seems to have come out of the blue at the very end of a long paper
Author Response
Comment 1:
This paper aims to review multiple facets/aspects of the manufacture and use bioengineered skin substitutes in Europe. It would be improved by significant editing to focus on the problem at hand, which is the introduction of legislation that takes no account of products in use for many decades in several centres, and is severely restrictive to manufacturing and translational research in this area. Following this, there needs to be a succinct and comprehensive set of structured proposals to address these problems.
Author's answer:
Thanks a lot, we are of the opinion that your concerns are largely valid and we have tried to improve the manuscript in accordance with your detailed suggestions. It must be noted that the challenge of such a paper is to combine the great expertise of the various experts, all of whom are also authors of this paper, in order to make the paper stringent in its argument. We have now tried to do this in the revised form, in line with your recommendation.
Comment 2:
Line 51 What is a ‘white paper’ in this context?
Author's answer:
Healthcare white papers are authoritative, in-depth documents that analyze specific medical, technical, or regulatory challenges while presenting research-backed solutions
Comment 3:
Line 63 – what does ‘long term efficacy’ refer to? It is well established that autologous bioengineered skin products persist and mature after engraftment. Perhaps consider expanding this statement for clarity
Author's answer:
Thank you for that suggestion. By long-term efficacy, we mean in particular the function of this skin over many decades; for example, we are conducting a study on this topic here in Switzerland, looking at patients who received autologous keratinocytes as part of an acute treatment 30-40 years ago.
Comment 4:
Section 2.1 the authors could consider whether the outlines of manufacturing methods for CEAs usefully add anything to the paper. Same should be considered for the autologous 2 layer substitutes, unless the methods details are not available elsewhere.
Author's answer:
We understand this very well. However, we felt it was important to give the reader an overview of all the methods that are currently relevant in clinical practice. We have, however, taken this suggestion on board and tried to make this section more balanced.
Comment 5:
Possibly a synthesis summarising the different products, and highlighting their differences and what the clinical and regulatory significance of these might be, would be more informative.
Author's answer:
We have tried to include this in the revision, thank you for pointing this out. Also look at the new table provided on page 10.
Comment 7:
Line 211 what does ‘cells are passed’ mean?
Author's answer:
Cells in culture, such as keratinocytes, stop their proliferation when they sense neighboring cells, ie confluence inhibition. In applications, such as the one descriped in the manuscript, you initially need to reach a high number of cells. Thus the keratinocytes need to be ‘diluted’, ie detached and moved/dispersed to culture flasks with less cells in order to keep proliferating. This is, in the lab world, refereed to as passing the cells (from one culture vessel to another). The number of passages is an indicator on ‘how old’ cultured cells are.
Comment 7:
Line 231: ‘might be at the cost…’ does not seem a very scientific way to describe anything
Author's answer:
Thank you very much we changed “If compared to the production of CEAs, the spray-strategy demands less manual labour and thus reduces the costs of production, although this might be at the cost of achieving lower take-rates after application” into: “If compared to the production of CEAs, the spray-strategy demands less manual labour and thus reduces the costs of production, however, lower take-rates is often observed”
Comment 8:
Line 270: elsewhere the locations of various institutes have been stated: where is the Tissue Biology Research Unit?
Author's answer:
The Tissue Biology Research Unit is not a manufacturing laboratory in the strict sense of the word (GCP) and was responsible for basic research and pre-clinical studies.
Comment 9:
Line 297; while co-operation between ICU and plastic surgery (assuming plastic surgeons are the burns surgeons) is relevant, it might be better to put the consideration of ‘bridging strategies’’ here with a more detailed consideration of the key role wound preparation plays in the success of bioengineered skin substitutes.
Author's answer:
We agree with this, an added: It is important to note that all of the methods described below require a bridging strategy. Ideally, allografts or dermal tempates are used to provide competent wound coverage after very early excision of the necrotic tissue until the autologous skin substitute is available.
Comment 10:
(Section 3.4 – use of ‘you’ is not scientific writing)
Author's answer:
Thank you, we've changed this. If different laboratory-produced autologous skin substitutes are used, a precise schedule is required.
Comment 11:
Section 3.1 ‘Indication’ tends to be a clinical term, although local protocols for using these products might be determined by availability.
Author's answer:
We have taken this into account in the text, thank you for pointing this out.
Comment 12:
Line 319 – not clear what size of burn has to do with a change to use in conjunction with ssg
Author's answer:
Thank you very much, we include this into the Table
Comment 13: Line 323 - this statement should have a reference
Author's answer:
Abdel-Sayed P, Michetti M, Scaletta C, Flahaut M, Hirt-Burri N, de Buys Roessingh A, Raffoul W, Applegate LA. Cell therapies for skin regeneration: an overview of 40 years of experience in burn units. Swiss Med Wkly. 2019 May 19;149:w20079. doi: 10.4414/smw.2019.20079. PMID: 31104308.
Comment 14:
Line 334 – reference for use and effectiveness with Meek and increased speed of healing
Author's answer:
Thank you very much, we added to the references:
- Burns Volume 36, Issue 3, May 2010, Pages e10-e20. Sprayed cultured autologous keratinocytes used alone or in combination with meshed autografts to accelerate wound closure in difficult-to-heal burns patients. S. Elizabeth James. Simon Booth, Baljit Dheansa, Dawn J. Mann, Michael J. Reid, Rostislav V. Shevchenko, Philip M. Gilbert
- Burns Volume 39, Issue 4, June 2013, Pages 674-679. The use of the Meek technique in conjunction with cultured epithelial autograft in the management of major paediatric burns. Seema Menon, Zhe Li, John G. Harvey, Andrew J.A. Holland
- Ann Burns Fire Disasters. 2020 Jun 30;33(2):134–142. Sprayed cultured autologous keratinocytes in the treatment of severe burns: a retrospective matched cohort study M Karlsson 1,∗, I Steinvall 1, P Olofsson 1, J Thorfinn 1, F Sjöberg 1, L Åstrand 2, S Fayiz 2, A Khalaf 2, P Divyasree 2, AT El-Serafi 1,2,3, M Elmasry 1
Comment 15:
Line 339 – 2- layer skin substitutes – what is meant by ‘established’? CEA and cell sprays are by no means standard of care in many/most burns services
Author's answer:
Established' here means that a large number of centres have access to this form of therapy. I do not think it is necessary to emphasise here that this is also the case for the other autologous skin substitutes, as we have mentioned this in other parts of the manuscript.
Comment 16:
Line 374: this is the first time ATMP is spelt out but it occurs before this in the text
Author's answer:
Thank you very much. You are right and we have changed it accordingly.
Comment 17:
Section 4 Regulatory issues: this section (especially 4.1) should be in the background/introduction, as it really contains the justification of the need for this paper
Author's answer:
Thank you very much. You are right and we have changed it accordingly.
Comment 18: Line 431: what does ‘diverse’ mean here and what is the significance of being evaluated on a case-by-case basis?
Author's answer:
What we mean is that the requirements for non-clinical data have changed in recent years according to the scientific knowledge of the ATMP products. For this reason, various EMA guidelines have been issued to provide recommendations for the characterisation of the product type. However, the revolution in the variety of advanced therapy products is a challenge not only for scientists but also for regulators, so the assessment of the associated risks needs to be evaluated on a case-by-case basis according to the intended use. We have included some clarification accordingly
Comment 19:
Line 440: why have hospitals stopped using the hospital exemption framework – does this exempt them from the GMP requirements?
Author's answer:
Thank you, we added: Despite the intention behind the 2009 regulations, there has been no tangible progress in the use of autologous skin substitutes for the treatment of severe burns. As a result, many laboratories have stopped providing manufacturing services under the hospital exemption, as manufacturing under the new regulatory requirements have driven up costs enormously.
Comment 20:
Section 5.2 Perhaps consider the availability of prefabricated GMP clean rooms and their potential to decrease costs?
Author's answer:
Thank you very much for making this valuable point, which we have avoided addressing for political reasons, but which should be one of the key points when negotiating with e.g. the EMA.
Comment 20:
Circling back to talk about another product does not seem to add much to what is already a very broadly focussed paper.
Author's answer:
We are sorry, but there seems to be a misunderstanding. PV-Skin is not an additional product, but rather a significant further development of the existing products.
Comment 21:
Line 525 ESEN and other abbreviations should probably be spelt out the first time they occur, even though there is an abbreviation section at the end.
Author's answer:
Thank you very much. You are right and we have changed it accordingly.
Comment 22:
Line 545: ‘Cost action’ seems to have come out of the blue at the very end of a long paper
Author's answer:
Thank you very much. You are right and we have changed it accordingly: A major success in the course of ESEN's active collaboration was that the European Union approved an application for support and ESEN became a Cost Action project in 2024.
Round 2
Reviewer 1 Report
Comments and Suggestions for Authors
The authors have provided a thorough revision. This should be a useful reference on a complex topic. There are a few punctuation and grammar errors, suggest a review from that standpoint please.
Reviewer 2 Report
Comments and Suggestions for Authors
The manuscript has been adequately revised, thank you.